# The Effects of an Artificial Garden on Heart Rate Variability among Healthy Young Japanese Adults

**DOI:** 10.3390/ijerph17249465

**Published:** 2020-12-17

**Authors:** Hiromi Suenaga, Kanako Murakami, Nozomi Murata, Syoriki Nishikawa, Masae Tsutsumi, Hiroshi Nogaki

**Affiliations:** Graduate School of Medicine, Faculty of Health Sciences, Yamaguchi University, Yamaguchi 753-8511, Japan; sunflower333@outlook.jp (K.M.); nzm_r_k4@icloud.com (N.M.); b010up@yamaguchi-u.ac.jp (S.N.); tutumi@yamaguchi-u.ac.jp (M.T.); nogaki@yamaguchi-u.ac.jp (H.N.)

**Keywords:** heart rate variability, artificial gardens, physiological effects, sympathetic nervous system, parasympathetic nervous system, nature environments

## Abstract

Spending time in nature might positively influence mental health by inducing a relaxed state. Recently, gardens have been created on hospital rooftops in Japan to help inpatients recover from various physical and mental aliments. However, there is little evidence regarding any positive physiological effects of artificial gardens designed for health. The purpose of this study was to assess the psychological and cardiovascular responses incited by artificial natural environments. Japanese university students (*n* = 38) participated in a one-group pretest post-test experiment conducted at the Yamaguchi Flower Expo in Japan in October 2018, designed to assess whether exposure to four environments (forest, flowers, ocean, and artificial garden for health) influenced heart rate variability measures. After pretesting to determine baseline measurements, participants completed a circuit through the four natural environments. Following circuit completion, post-testing determined that the low frequency/high frequency ratio was significantly lower in the overall sample and the four areas had similar influences on heart rate variability. Findings suggest that exposure to nature by walking through natural areas and in rooftop artificial gardens might enhance the balance between the sympathetic and parasympathetic nervous systems.

## 1. Introduction

Mental health problems, such as depression, are common and pose serious health problems. According to the 2004 World Health Organization’s World Mental Health Surveys, the prevalence of mental health problems in the United States was high in (26.4%) [1]. In urban or stressful societies, some psychological problems are increasingly influencing health [2,3,4,5,6,7]. Some studies have found mental health benefits associated with contact with nature [8], and the stress-reducing influence of visiting public parks is well established [9]. Recent experimental research has confirmed those effects [10], and large-scale epidemiological studies, particularly on green space and health, reported that forest bathing or forest therapy improved physical and mental health because of the forest aspect of the experience [11,12,13,14]. Previous studies on mental health and occupational health found that images of nature, such as forests [15], oceans [16], or flowers [17,18] promoted relaxation. Apparently, forests’ therapeutic effects are caused by the forest environment’s activation of parasympathetic nervous activity and improvement of immune function resulting from the inhalation of D-limonene and α-pinene, which are major components of forest environments [19]. Just looking at pictures or videos of forests has enhanced physiological [20] and psychological [21] relaxation. Many studies have examined the relationships between green space, including forests and gardens, and health, but only a few studies have been conducted on blue spaces (visible surface waters, i.e., lakes, rivers, and oceans), and due to methodological heterogeneity, the association with health remains unclear [22,23,24]. In addition, physical activity in a natural environment has been reported to benefit mental health compared to indoor physical activity [25]. Other similar studies have also demonstrated that short-term exposure, like a 15 min walk in green space, could have health benefits [26,27]. To apply these findings, some modern Japanese hospitals have built rooftop gardens that resemble natural greenery so that inpatients can relax and recover. However, there is little research on the psychological and physical effects of artificial gardens on inpatient health. In addition, the differences in the influences of different types of nature, such as forest, flower, and the ocean, on individual health are unclear.

Therefore, this study investigated whether an artificial garden designed for health influenced health in ways similar other types of nature. Data were objectively obtained by continuously recording the heartrate variability (HRV) of a sample of 38 Japanese university students while they walked through four human-created environments (forest, flower, ocean, and Garden for Health).

## 2. Materials and Methods

### 2.1. Participants

Twenty-five female and 13 male Japanese university students (Age M = 21.7, SD = 1.1) were recruited to participate in this study. Emails were sent and posters were displayed on campus to attract students to participate. The inclusion criteria were: (1) non-smoker; (2) not pregnant; (3) no history of cardiac disease (e.g., arrhythmia); (4) not currently taking medication for hypertension, depression, anxiety, or sleep; and (5) ability to walk for 30 min without a break. All the participants were fully informed about the study’s purposes and procedures. By signing the consent form, they agreed to participate in the study.

This study was conducted in accordance with the Declaration of Helsinki, and the Ethics Committees of the Center for Environment, Yamaguchi University Graduate School of Medicine, Japan (project identification no. 548), approved its protocol.

### 2.2. Design and Procedure

#### 2.2.1. Study Site

This study was conducted at the Yamaguchi Flower Expo in Japan in October 2018. Four areas of the expo were used to gather the participants’ HRV data: (1) 3.6 ha planted forest, (2) 3.6 ha planted flower area, (3) 3.9 ha ocean area, and (4) 043 ha Garden for Health (Figure 1 and Figure 2). The Japan Landscaping Association and Yamaguchi University Hospital conducted these exhibitions for health promotion.

#### 2.2.2. Preliminary Survey

The preliminary survey was conducted a week before the participants were exposed to the treatment (the walk through the four areas of the study site). They were asked about three physical conditions that might be associated with their participation: allergic rhinitis, sleep disorder, and regular exercise habits. Allergic rhinitis was added to the questionnaire due to concerns that walks in the flower or forest areas might result in allergic rhinitis. At the same time, the participants underwent evaluation of autonomic nervous functions as baseline HRV in the sitting position using the Fatigue Measurement Device VM302 (Hitachi Systems Co. Ltd., Tokyo, Japan) in the university laboratory.

The VM302 evaluates autonomic nervous function by monitoring HRV in 90 s electrocardiogram (ECG) interval data, which are transmitted to an external computer.

The collected R-R interval variation from each R peak time was analyzed using Mem Calc/Win (GMS Co., Ltd., Tokyo, Japan) with frequency analysis performed with the maximum entropy method (MEM). Consequently, HRV indicators were automatically calculated regarding the power levels of the high-frequency zone (HF: 0.15 Hz–0.40 Hz) and the low-frequency zone (LF: 0.04 Hz–0.15 Hz). HF power indicates parasympathetic nervous activity, and the LF/HF power ratio indicates the balance between the sympathetic and parasympathetic systems [28]. Hitachi Systems Co. Ltd. uses a standard LF/HF power ratio of 0.8 Hz–2.0 Hz; values below 0.8 Hz are considered a hyper-sympathetic state and values above 2.0 Hz are considered a sympathetic state. The ccvTP corrected the sum of LF and HF by heartrate and was used as an index of the magnitude of the total autonomic nervous system’s activity. The ccvTP was also used as a reference value in this study because it is automatically analyzed by the Fatigue Measurement Device VM302 to produce autonomic activity.

#### 2.2.3. Experimental Procedure

The experiment was conducted at the Yamaguchi Flower Expo between 9 AM and 11 AM in October 2018. This study was a pretest post-test quasi-experiment. All participants were asked to abstain from alcohol consumption inoculation 12 h before the measurement and from food eating and caffeine consumption 1 h before. In the pretest before the walk, all the participants’ autonomic nervous functions were recorded in the sitting position in the Garden of Health using the Fatigue Measurement Device VM302 to set the starting point and baseline of the experiment. The procedure was identical to the procedure performed during the preliminary survey. Then, all the participants were fitted with the Holter ECG RAC-2512 (Nihon Koden, Tokyo, Japan) to continuously record HRV during the experiment. All the ECG data were directly transmitted to a computer as R-R intervals via the Holter ECG software. The excerpts of the fatigue stress instrument measured before and after walking were ultra- short term, less than 5 min, while the excerpt on the Holter ECG during walking was short-term, between 30 min and 1.5 h [28].

All participants carried GPS loggers that matched ECG data to their locations in the walking circuit, and they were allowed to freely move around the study site. The GPS loggers recorded their locations as longitude and latitude, and the locations were identified as one of the four areas using these data.

To assess the influences of exposure to the four areas, the HRV values were computed and analyzed using maximum entropy method software (Mem Calc, GMS, Tokyo, Japan). The coefficient of variance of the R-R intervals (CVRR: standard deviation of the R-R interval/mean of the R-R interval × 100) was calculated to obtain a value representing total HRV power.

The study site used a circuit designed so that visitors entered at the Garden for Health; walked through the forest, flower, and ocean areas; and reached the end of the circuit at the Garden for health.

Once they completed the circuit and were again in the Garden for Health, each participant’s autonomic nervous functions were assessed in the sitting position using a fatigue measurement device, VM302.

Lastly, participants were asked whether they thought the crowd was uncomfortably large and which of the four areas was their favorite area.

### 2.3. Data Management and Statistical Analysis

The mean HRV values of each participant during exposure to each of the four areas were calculated. Paired *t*-tests were performed to compare the pretest to the post-test means.

A one-way repeated measures analysis of variance (ANOVA) test was performed to test differences in the HRV mean values of the four areas and post hoc Tukey Tests were used for the multiple comparisons procedure. The cut-off value of statistical significance on all tests was *p* < 0.05. All statistical analyses were performed using JMP Pro 13 (SAS Institute Inc, Charlotte, NC, USA).

## 3. Results

The participants’ information and characteristics are presented in Table 1. The pretest data indicated that approximately 39% (*n* = 15) of the participants reported allergic rhinitis, 13% (*n* = 5) reported sleep disorders, and approximately about 29% (*n* = 11) reported regular physical exercise.

### 3.1. Baseline HRV

First, independent samples *t*-tests were used to compare mean HRV scores between those who reported and those who did not report allergic rhinitis, sleep disorder, or regular physical exercise.

Table 2 indicates a significantly lower HF among those who regularly engage in physical exercise than those who do not, and their LF/HF and heartrate were significantly lower than those without regular physical exercise. Most of those who practiced regular physical activity were in the normal range as assumed by the equipment manufacturer. The mean LF/HF of those with sleep disorders was significantly higher than those without a disorder, and most of with sleep disorders were above normal range. Mean HF among those with allergic rhinitis was higher than that of those without it, suggesting that allergic rhinitis might enhance parasympathetic nervous activity, although the difference was not statistically significant.

### 3.2. HRV Change after the Walk

The HVR was significantly different after the walk only regarding LF/HF (*p* = 0.08) (Table 3).

### 3.3. HRV Changes in Four Areas (Forest, Flower, Ocean, and Garden of Health)

Neither the HF nor the LF/HF was significantly different after the walk, but CVRR tended to be lower in all areas compared to before the walk. The amount of time spent in each area ranged from 2–60 min (Table 4). In particular, LF/HF after spending more than 10 min in the Garden of Health (*n* = 16, mean = 1.7 ± 1.9) was significantly lower than among those who spent less than 10 min there (*n* = 20, mean = 4.1 ± 4.0, *p* = 0.04).

The participants’ HRV differences after the walk by condition (allergic rhinitis, sleep disorder, physical exercise) are shown in Table 5.

HF in 10 of the 15 participants who reported allergic rhinitis was lower after the walk.

LF/HF in among those who did not regularly exercise, which were often increased, tended to decrease and improve after walking.

### 3.4. Changes in LF/HF after the Walk

Figure 3 indicates that the participants with low LF/HF scores (below the reference range of 0.8–2.0) had significantly higher LF/HF scores after the walk and higher sympathetic nervous activity. In contrast, 73% of the 15 participants with high LF/HF in the pretest had lower LF/HF after the walk and a tendency toward reduced sympathetic hyperactivity.

LF/HF scores higher than 2.0 were lower after the walk (*n* = 15, *p* = 0.13), whereas LF/HF scores less than 0.8 were significantly higher after the walk (*n* = 9, *p* < 0.05).

## 4. Discussion

Several systematic reviews have indicated that natural environments, such as forests, influenced physical health and relaxation. In medicine, flowers in hospital rooms might reduce patients’ pain and stress. Rooftop gardens built at hospitals aim to promote patients’ psychological recovery by providing comfort. However, there are no studies comparing the physiological effects among types of environments, such as artificial gardens, forest, ocean, and flower settings.

The Yamaguchi Flower Expo was selected as the experimental site because it had a Garden of Health designed for hospitals along with forest, flower, and ocean areas.

This study aimed to assess the effects of exposure to an artificial garden on the autonomic nervous system’s activity by comparing its effects to those of the other natural environments at the Expo.

Sympathetic and parasympathetic nervous system activity was measured using HRV.

The HF, heartrate, and ccvTP values were not significantly different after the walk, but the LF/HF was significantly lower than before the walk, suggesting that exposure to the natural surroundings might have had a suppressing influence on sympathetic nervous activity. Nearly one-quarter (*n* = 3, 23%) of the men and almost one-half (*n* = 12, 47%) of the women indicated that they had allergic rhinitis. Some previous studies found that allergic rhinitis sufferers were predisposed to parasympathetic nervous activity [29,30], many of which found that parasympathetic nervous activity among allergic rhinitis sufferers was suppressed by walking, resulting in an improved balance in autonomic nervous activity.

The present study supports these findings because the participants without allergic rhinitis had significantly lower HF after the walk (510 and 293, respectively, *p* < 0.05), but the effect was not significant for those with allergic rhinitis.

Most of the participants with sleep disorders (*n* = 5) were markedly hyper sympathetic, and the LF/HF power ratio indicated that balance between the sympathetic and parasympathetic activity tended toward sympathetic nervous activity. The LF/HF and the heartrate were significantly higher after the walk (*p* < 0.05) among those without a sleep disorder (*n* = 33), suggesting that the walk increased parasympathetic nervous activity and improved the balance between the sympathetic and parasympathetic activity. Those with low LF/HF before the walk had higher LF/HF after the walk, indicating improved sympathetic activity, whereas those with high LF/HF before the walk had lower LF/HF after the walk, indicating reduced sympathetic hyperactivity. These findings suggest that walking influenced and regulated the balance of the autonomic nervous activity.

The CVRR was used to measure total HRV power.

The differences after the walk were not statistically significant, but both measures were higher before than after the walk, and they gradually declined during the walk. The participants who reported that they were not physically active (*n* = 28) had significantly lower HF and ccvTP after the walk and a significantly higher heartrate after the walk, but the changes in HRV were non-significant among those who reported physical activity (*n* = 10). These findings suggest that fatigue after a 30–60 min walk might reduce autonomic nervous activity and more strongly influence people who do not regularly engage in physical exercise.

## 5. Conclusions

In this study, the participants were exposed to four types of nature (artificial garden, forest, flowers, and ocean), and no significant differences were found in their effects on the autonomic nervous system. These findings suggest that artificial gardens for health promotion have the same influence as the forest [15], flower [18], and ocean [16], all of which were previously found to have positive effects (e.g., relaxation) on the autonomic nervous system.

Additionally, more than 10 min of exposure to an artificial garden might be more effective than less exposure. Hospital gardens might enhance their effectiveness for patients’ psychological health by providing tools inpatients can use to extend their stays there.

However, this study has some limitations. A major limitation was that the site of the experiment was the Expo site with an artificial garden, which would be difficult to reproduce. The participants were young university students, but many of the patients who use hospital gardens are middle-aged or older, so this experiment should be replicated on samples of older individuals. The effects of variation in the ambient temperature on the autonomic nervous system could not be assessed because the experiment was conducted only during the summer. Depending on the free walk, the walking route and time differed by individual. The differences in the time spent in each area varied because the participants controlled the walk, and the temporal differences among the areas might have influenced the results; however, this study did not test those differences.

Lastly, the large number of visitors at the study site might have congregated as crowds of people that excited the participants in ways that influenced their results. Despite these caveats, this study found that exposure to the Garden of Health and the other natural areas had a beneficial influence on balancing the autonomic nervous systems of healthy young adults.

## Figures and Tables

**Figure 1 ijerph-17-09465-f001:**
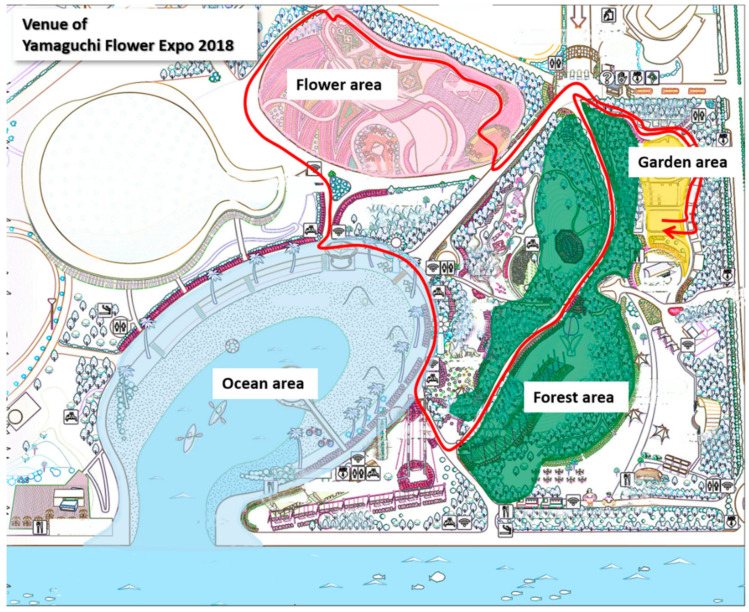
Study site: flower area (pink), ocean area (blue), forest area (green), and Garden for Health (yellow). The red trails indicate the main walking circuit.

**Figure 2 ijerph-17-09465-f002:**
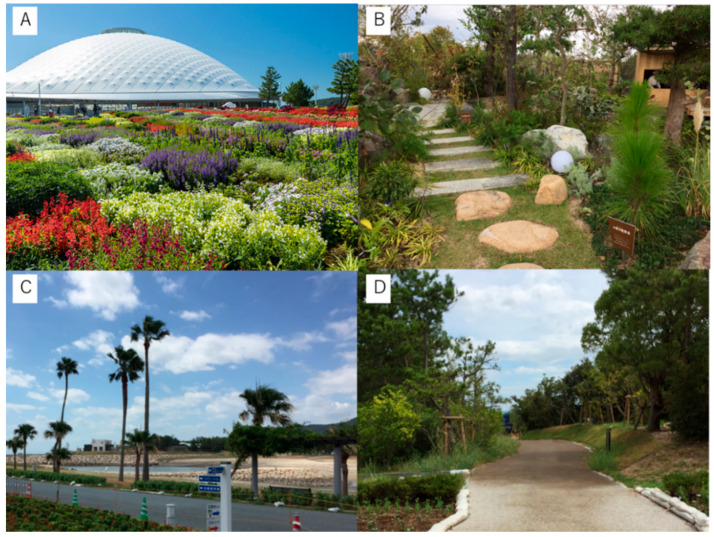
(**A**): Image of flower area, (**B**): Image of Garden for Health, (**C**): Image of ocean area, (**D**): Image of forest area.

**Figure 3 ijerph-17-09465-f003:**
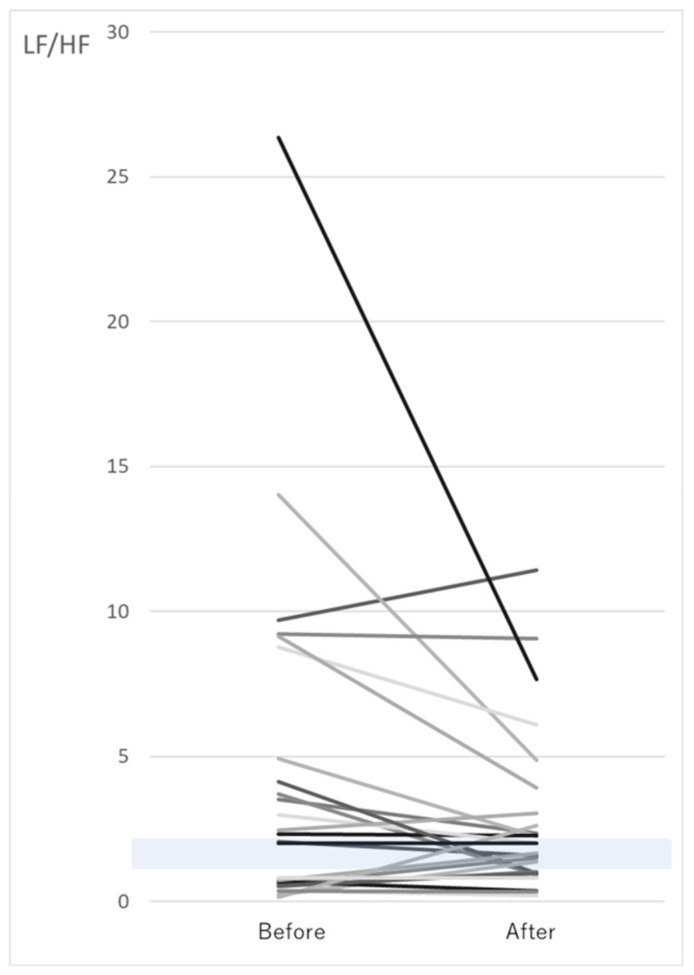
Changes in LF/HF; the gray zone indicates the LF/HF reference range of 0.8–2.0. Different colored lines indicate individual subjects.

**Table 1 ijerph-17-09465-t001:** Participants’ descriptive statistics (*n* = 38).

Variable	Males (*n* = 13)	Females (*n* = 25)	All (*n* = 38)
	Mean (SD)	Mean (SD)	Mean (SD)
Age (years)	21.9 (1.0)	21.2 (1.2)	
Height (cm)	170.2 (6.8)	158.2 (5.3)	
Weight (kg)	62.0 (8.0)	49.8 (6.0)	
BMI (kg/m^2^)	21.2 (1.9)	19.9 (2.2)	
	**Males (*n* = 13)**	**Females (*n* = 25)**	**All (*n* = 38)**
**Physical symptom**	***n* (%)**	***n* (%)**	***n* (%)**
allergic rhinitis	3 (23)	12 (48)	15 (39)
sleep disorder	2 (15)	3 (12)	5 (13)
fitness habits	3 (23)	8 (32)	11 (29)

BMI: Body Mass Index.

**Table 2 ijerph-17-09465-t002:** Baseline heartrate variability data in the overall sample and by health condition; statistical significance is of participant’ *t*-test results (*n* = 38).

Variable		Allergic Rhinitis			Sleep Disorder			Fitness Habits		
	All (*n* = 38)	Presence (*n* = 15)	Absence (*n* = 23)	*p*-Value	Presence (*n* = 5)	Absence (*n* = 33)	*p*-Value	Presence (*n* = 10)	Absence (*n* = 28)	*p*-Value
	Mean (SD)	Mean (SD)	Mean (SD)		Mean (SD)	Mean (SD)		Mean (SD)	Mean (SD)	
HF (ms^2^)	721 (695)	937 (897)	585 (462)	0.13	508 (298)	752 (720)	0.48	1089 (803)	576 (572)	<0.05
LF/HF	1.7 (2.0)	1.2 (1.0)	1.9 (2.4)	0.26	3.6 (3.5)	1.4 (1.5)	<0.05	0.7 (0.6)	2.0 (2.3)	0.08
HR (bpm)	75.1 (11.8)	72.1 (3.0)	77.3 (2.5)	0.19	79.8 (5.3)	74.5 (2.1)	0.36	69.1 (3.4)	77.7 (2.2)	<0.05
ccvTP	4.1 (1.4)	4.4 (1.3)	3.9 (1.4)	0.27	4.0 (1.3)	4.2 (2.0)	0.31	4.4 (1.0)	3.9 (1.5)	0.34

HF: high frequency; LF/HF: low frequency/high frequency ratio; HR: heart rate; ccvTP: coefficient of component variance total power.

**Table 3 ijerph-17-09465-t003:** Mean heart rate variability values pretest and post-test (*n* = 38).

Variable	Pretest	Post-Test	*p*-Value
	Mean (SD)	Mean (SD)	
HF (ms^2^)	557 (619)	451 (619)	0.20
LF/HF	3.4 (4.9)	3.3 (3.4)	0.08
HR (bpm)	75.7 (10.1)	76.1 (9.2)	0.84
ccvTP	4.2 (1.5)	3.8 (1.5)	0.58

HF: high frequency; LF/HF: low frequency/high frequency ratio, HR: heart rate; ccvTP: coefficient of component variance total power.

**Table 4 ijerph-17-09465-t004:** Mean HRV by environmental area.

Variable	Garden for Health	Ccean	Flowers	Forest	ANOVA	
	Mean (SD)	Mean (SD)	Mean (SD)	Mean (SD)	F-Value	*p*-Value
HF (ms^2^)	145 (130)	125 (109)	125 (109)	123 (104)	0.31	0.81
LF/HF	6.7 (3.7)	6.6 (3.5)	6.9 (3.5)	6.2 (3.0)	0.25	0.86
HR (bpm)	89.4 (12.3)	93.4 (9.7)	91.4 (10.3)	93.1 (12.5)	0.91	0.44
CVRR	5.0 (1.5)	5.0 (1.1)	5.0 (1.3)	4.8 (1.4)	0.32	0.81
log in time (min)	3–35	2–14	7–28	8–60		

HF: high frequency; LF/HF: low frequency/high frequency ratio; HR: heart rate; CVRR: coefficient of variation R-R interval.

**Table 5 ijerph-17-09465-t005:** Mean pretest and post-test heart rate variation by condition, allergic rhinitis, sleep disorder and fitness habits.

Variable		Allergic Rhinitis			Sleep Disoder			Fitness Habits		
		Presence (*n* = 15)	Absence (*n* = 23)	*p*-Value	Presence (*n* = 5)	Absence (*n* = 33)	*p*-Value	Presence (*n* = 10)	Absence (*n* = 28)	*p*-Value
HF (ms^2^)	pretest	896 (835)	510 (898)	0.18	309 (396)	710 (152)	0.35	1184 (1210)	452 (633)	<0.05
	post-test	770 (895)	293 (309)	<0.05	445 (395)	481 (672)	0.91	1099 (983)	232 (186)	<0.0001
LF/HF	pretest	2.2 (3.1)	4.1 (5.7)	0.24	9.5 (10.3)	2.5 (3.0)	<0.01	2.1 (1.5)	3.8 (0.9)	0.34
	post-test	2.5 (3.2)	3.6 (3.6)	0.33	5.0 (3.2)	3.0 (3.4)	0.21	2.4 (3.4)	3.5 (3.5)	0.39
HR (bpm)	pretest	71.9 (8.0)	78.1 (10.7)	0.06	84.8 (6.1)	74.4 (9.9)	<0.05	69.5 (8.3)	78.1 (9.8)	<0.05
	post-test	72.7 (7.8)	78.3 (9.6)	0.07	80.9 (7.0)	75.4 (9.4)	0.22	68.8 (7.3)	79.0 (8.4)	<0.001
ccvTP	pretest	4.7 (0.5)	4.1 (0.4)	0.35	4.7 (0.9)	4.3 (0.3)	0.66	5.3 (2.4)	4.0 (1.5)	<0.05
	post-test	4.2 (1.7)	3.7 (1.4)	0.29	5.0 (1.8)	3.7 (1.5)	0.08	5.1 (1.5)	3.4 (1.3)	<0.01

HF: high frequency; LF/HF: low frequency/high frequency rati; HR: heart rate; ccvTP: coefficient of component variance total power.

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
