# Peer review of "The Effects of an Artificial Garden on Heart Rate Variability among Healthy Young Japanese Adults"

_ijerph, 2020, doi:10.3390/ijerph17249465_

Round 1

Reviewer 1 Report

The manuscript ijerph-992857, Title: The Effects of an Artificial Garden on Heart rate Variability among Healthy Young Japanese Adults, tries to assess whether exposure to four environments (forest, flowers, ocean, 13 and artificial garden for health) influenced heart rate variability measures.

One major issue of this study is the very low participation rate and age of the participants, limiting generalizability and representativeness of the study. The high amount of subjects with hay fever symptoms could be an indicator for selection bias. Also, the recruitment process is unclear. What time period is spanned in “on October, 2018.” (sic!)? The English language needs to be checked by a native speaker.

Comments:

The abstract seems to be too long and not structured according to the journal`s requirements.

The intro is very short and would profit from a more structured content including a clear research hypothesis.

Is there any information on medication intake, especially among participants with physical symptoms? The rationale for hay fever and effects of walking is not adequately described and the terms hay fever and allergic rhinitis are inconsistently used, evoking irritation.

The results should not start with table 1 right away.

The tables show different styles and most of the headings need to be rephrased.

The information provided of Figure 1. is unclear. what do these lines mean?

When stating “Several systematic reviews”, those should be cited.

A more detailed limitation section is needed, as generalizability and representativeness of the study is not given.

Illustration of the paths walked and pictures of the different places are missing.

Information on Author Contributions, Funding, and Conflicts of Interest is missing.

The literature section is mostly outdated and more recent papers on recreation in outdoor nature are missing.

Author Response

The abstract seems to be too long and not structured according to the journal`s requirements.

The intro is very short and would profit from a more structured content including a clear research hypothesis.

Is there any information on medication intake, especially among participants with physical symptoms? The rationale for hay fever and effects of walking is not adequately described and the terms hay fever and allergic rhinitis are inconsistently used, evoking irritation.

  Answer: Thank you for your careful review and important suggestions. In response to your suggestions, we have revised the Abstract, added missing exclusion criteria such as medication, and ensured that allergic rhinitis is consistently used in place of hay fever. We have added why participants asked about their history of allergic rhinitis(P4.166-167)., and added items such as medication to the exclusion criteria (P2.112-113). And we have unified with allergic rhinitis rather than hay fever.

In addition, we have clarified how participants were recruited for the study.

The results should not start with table 1 right away.

Answer: In response to your suggestion, we have corrected the placement of Table 1. The tables show different styles and most of the headings need to be rephrased.

We attempted to make the table headings consistent.

The information provided of Figure 1. is unclear. what do these lines mean?

   Answer: In Figure 2, the LF/HF of each subject was connected by lines to match before and after. In addition, the gray area illustrates the normal reference values generated by the fatigue stress measurement device.

A more detailed limitation section is needed, as generalizability and representativeness of the study is not given.

Answer: We have added the limitations of this study in the Conclusions section.

Illustration of the paths walked and pictures of the different places are missing.

Answer: We have added an illustration of the walking paths and photographs of the four natural environment areas.

Information on Author Contributions, Funding, and Conflicts of Interest is missing.

The literature section is mostly outdated and more recent papers on recreation in outdoor nature are missing.

Answer: We have added a description of recent literature about the physical and psychological benefits of walking and blue spaces in the introduction.

Reviewer 2 Report

I have a few observations/questions to authors.

What about coffee or another stimulants consumption? It had an influence on participants.

You said that participants reported fever…. Are they really go to the forest with fever?

Table 3. How post-pre significance was calculated? In which statistical method?

I have a problem with statistical method – maybe repeated measure ANOVA should be applied?

What is “sojoum time”? (Table 4).

Maybe some photos of these places might be shown in the manuscript?

Maybe exceptional map of travel of participants might be presented in the manuscript?

Author Response

What about coffee or another stimulants consumption? It had an influence on participants.

Answer: Thank you for your careful review and important suggestions. Additional exclusion criteria were added for coffee consumption and medications that were not previously described.

You said that participants reported fever…. Are they really go to the forest with fever?

 Answer: Despite our misleading representations, none of the participants showed symptoms or a fever on the day of the experiment, although some reported a history of allergic rhinitis.

Table 3. How post-pre significance was calculated? In which statistical method?

I have a problem with statistical method – maybe repeated measure ANOVA should be applied?

Answer: Table 3 shows statistical significance using the paired t-test. In addition, we have described the statistical methods used for the study.

What is “sojoum time”? (Table 4).

Answer: The erroneous “sojoum time” in Table 4 has been corrected to "log in time."

Maybe some photos of these places might be shown in the manuscript?

Maybe exceptional map of travel of participants might be presented in the manuscript?

Answer: We have added an illustration of the walking paths and photographs of the four areas.

Reviewer 3 Report

Thanks for the opportunity to review the manuscript "The Effects of an Artificial Garden on Heart rate Variability among Healthy Young Japanese Adults", which I find fascinating and timely in terms of both research question and methods used. I do have some concerns that I list below in no particular order aiming to move the contribution forward, mostly on the methods

  • I would like some clarity on the rationale for using the ccvTP corrected approach
  • how long did the circuit last in total within the 2 hours?
  • how long were the excerpt analyzed? short or long term? ie at least 5-minutes? see castaldo et al 2017
  • how do you interpret LF/HF in light of sympathovagal balance information in terms of behavioral research? see massaro and pecchia 2019 also for useful guidelines in hrv research

good luck in advancing your work

Massaro, S., & Pecchia, L. (2019). Heart rate variability (HRV) analysis: A methodology for organizational neuroscience. Organizational Research Methods22(1), 354-393.

Author Response

I would like some clarity on the rationale for using the ccvTP corrected approach

Answer: Thank you for your careful review and important suggestions. Because the fatigue stress measurement device used during the study automatically calculates A as autonomic activity, we also evaluated it as a reference value.

how long did the circuit last in total within the 2 hours?

Answer: Walking times are listed in Table 4, however, the longest times were approximately 1 hour and 30 minutes.

how long were the excerpt analyzed? short or long term? ie at least 5-minutes? see castaldo et al 2017

Answer: The analyses were continuously recorded from before to after the walk.

And added the following sentence about excepts to P4.194.

“The excerpts of the fatigue stress instrument measured before and after walking were ultra- short term, less than 5 minutes, while the excerpt on the Holter ECG during walking was short-term, between 30 minutes and 1.5 hours.”

how do you interpret LF/HF in light of sympathovagal balance information in terms of behavioral research? see massaro and pecchia 2019 also for useful guidelines in hrv research good luck in advancing your work. Massaro, S., & Pecchia, L. (2019). Heart rate variability (HRV) analysis: A methodology for organizational neuroscience. Organizational Research Methods, 22(1), 354-393.

Answer: Similar to other researchers, we consider the LF/HF ratio to be an indicator of sympathetic and parasympathetic balance. Therefore, this study’s findings that walking decreased LF/HF in subjects with high LF/HF and sympathetic dominance and increased LF/HF in subjects with parasympathetic dominance and low LF/HF indicate that artificial gardens could have an effect on improving autonomic activity. Thank you for introducing an excellent review. I have cited it as a reference.

Round 2

Reviewer 1 Report

I thank the authors for their corrections. Still, some minor editing is needed.